# Endovascular Use of Cyanoacrylate-Lipiodol Mixture for Peripheral Embolization: Properties, Techniques, Pitfalls, and Applications

**DOI:** 10.3390/jcm10194320

**Published:** 2021-09-23

**Authors:** Pierre-Olivier Comby, Kévin Guillen, Olivier Chevallier, Marc Lenfant, Julie Pellegrinelli, Nicolas Falvo, Marco Midulla, Romaric Loffroy

**Affiliations:** 1Department of Neuroradiology and Emergency Radiology, François-Mitterrand University Hospital, 14 Rue Paul Gaffarel, BP 77908, 21079 Dijon, France; pierre-olivier.comby@chu-dijon.fr (P.-O.C.); marc.lenfant@chu-dijon.fr (M.L.); 2Imaging and Artificial Vision (ImViA) Laboratory-EA 7535, University of Bourgogne/Franche-Comté, 9 Avenue Alain Savary, BP 47870, 21078 Dijon, France; kguillen@hotmail.fr (K.G.); olivier.chevallier@chu-dijon.fr (O.C.); 3Department of Vascular and Interventional Radiology, Image-Guided Therapy Center, François-Mitterrand University Hospital, 14 Rue Paul Gaffarel, BP 77908, 21079 Dijon, France; julie.pellegrinelli@chu-dijon.fr (J.P.); nicolas.falvo@chu-dijon.fr (N.F.); marco.midulla@chu-dijon.fr (M.M.)

**Keywords:** interventional radiology, transcatheter embolization, arteries, veins, cyanoacrylate, glue

## Abstract

Endovascular embolization agents are particles and fluids that can be released into the bloodstream through a catheter to mechanically and/or biologically occlude a target vessel, either temporarily or permanently. Vascular embolization agents are available as solids, liquids, and suspensions. Although liquid adhesives (glues) have been used as embolic agents for decades, experience with them for peripheral applications is generally limited. Cyanoacrylates are the main liquid adhesives used for endovascular interventions and have a major role in managing vascular abnormalities, bleeding, and many vascular diseases. They can only be injected as a mixture with ethiodized oil, which provides radiopacity and modulates the polymerization rate. This review describes the characteristics, properties, mechanisms of action, modalities of use, and indications of the cyanoacrylate-Lipiodol^®^ combination for peripheral embolization.

## 1. Introduction

Embolic agents can be classified by their physical and biological properties, which affect the level of occlusion and tissue response. Most interventional radiologists are familiar with the use of mechanical and particulate agents. Conversely, experience with liquid embolic agents, notably cyanoacrylate glues, is generally far more limited. However, specific characteristics of cyanoacrylates confer high suitability to the management of a wide variety of vascular conditions affecting the peripheral circulation. These characteristics include high radiopacity when mixed with ethiodized oil (Lipiodol^®^ Ultra Fluid, Guerbet, Villepinte, France), rapid embolization, a low rate of same-vessel recanalization, and the capacity to penetrate the vascular bed in a flow-directed fashion. Some of the advantageous properties of cyanoacrylate glues may also create a risk of complications, although these are uncommon when embolizing peripheral sites using the proper technique.

This review discusses the properties, mechanisms of action, techniques of use, main indications, and potential pitfalls of endovascular cyanoacrylate-glue embolization at peripheral sites.

## 2. Characteristics

### 2.1. Biological Cyanoacrylate Glues

Glue was first used for embolization in 1975 [1]. Embolization with cyanoacrylate glue is very challenging, as the occlusion is permanent and almost instantaneous. Therefore, cyanoacrylate embolization should be reserved for specialized centers. The most widely used cyanoacrylate glue is now Histoacryl^®^ (B/Braun, Tuttlingen, Germany), i.e., *N*-butyl-2-cyanoacrylate (NBCA), which is also available as Trufill^®^ (Cordis, Miami Lakes, FL) and has superseded isobutyl cyanoacrylate (Figure 1). Glubran^®^2 (GEM SRL, Viareggio, Italy) is now also a well-known surgical glue in which NBCA is combined with another monomer, metacryloxysulfolane, to produce a more pliable polymer in which the milder exothermic reaction (45 °C) results in less histotoxicity and inflammation compared to Histoacryl^®^ and Trufill^®^ (Figure 1) [1,2]. The long-term fate of Glubran^®^2 glue in the body is unknown.

Few cyanoacrylates are officially available on the market worldwide for endovascular purpose: Glubran2^®^ and Trufill^®^ have the CE mark and FDA approval, respectively [3]. Histoacryl^®^ has neither the CE mark nor FDA approval for endovascular interventions, for which it is widely used off label. Its faster polymerization rate compared to other glues makes Histoacryl^®^ more challenging to use [4]. Glubran^®^2 has the advantages of being very inexpensive compared to Trufill^®^ (about 100 € versus 2000 $ per 1 mL vial) and available worldwide, whereas Trufill^®^ is used only in the United States. A new α-hexil-cyanoacrylate glue (MagicGlue^®^, Balt Extrusion, Montmorency, France), known as Purefill^®^ (Peters Surgical, Bobigny, France), was recently developed and has exhibited short- and long-term occlusive efficacy and histopathologic responses similar to those seen with Histoacryl^®^ and Glubran^®^2 [5]. However, α-hexil-cyanoacrylate seems to have less adhesive strength compared to Histoacryl^®^ and Glubran^®^2.

### 2.2. Chemical Properties of Cyanoacrylates

Cyanoacrylates are the main liquid embolic adhesives used for endovascular interventions. The monomeric form is an ethylene molecule with a cyano group and an ester group attached to one of the carbons (Figure 2). The ester group can have several hydrocarbons attached to it (at the R position). In this position, the hydrocarbon also contributes to the name of the cyanoacrylate glue, e.g., isobutyl cyanoacrylate, *N*-butyl cyanoacrylate (NBCA), or 2-hexyl cyanoacrylate.

When exposed to an anion, such as a hydroxyl moiety found in water or the different anions that we can find in blood, polymerization starts with the bonding of the ethylene units. A longer hydrocarbon at the R position results in a slower polymerization rate, less heat released during polymerization, and less histotoxicity [6,7]. Because of their low viscosity, cyanoacrylates are quite easy to inject through microcatheters; however, their lack of radiopacity and rapid polymerization upon contact with blood can make precise and safe occlusion difficult to achieve. Modifications available for addressing these problems include the addition of ethiodized oil (Lipiodol^®^) [8]. Ethiodized oil not only opacifies the material but also slows the polymerization rate and promotes the formation of a less uniform and more flocculent polymer [6]. Tantalum can also be added to obtain radiopacity. Because tantalum slows the initiation of polymerization, it should be added only shortly before use of the embolic mixture [9].

## 3. Effects on Tissues

The effects of glues on body tissues have been studied in detail. The deposition of cyanoacrylate glue within a vessel results in an acute inflammatory reaction in the wall and tissues around. This leads to a chronic and granulomatous process within approximately 30 days, with foreign-body giant cell accumulation and fibrosis [9,10,11]. The main effects of glue injection are listed below.

Necrosis of the intima;Perivascular edema due to the early toxicity of the exothermic reaction (which can lead to pain) and the delayed toxicity of cyanoacrylate monomers released by degradation or left free by incomplete polymerization; toxicity is greatest during the first few weeks;A foreign-body reaction with infiltrative reaction by macrophages, and, subsequently, by giant cells; fibrosis develops gradually, with incorporation of the material into the vessel wall and slow degradation of the material over a couple of years [9,10,11]. Histologically, extravascular extrusion of the glue and the development of capillaries within the embolized vessels have been described.

The main advantage of cyanoacrylates is the longer-lasting nature of the vascular occlusion as compared to particles. However, the material has slow resorption, and recanalization may occur occasionally [9,10,11,12,13]. Vessel remodeling can take place, as with particles, leading in very few cases to glue migration in the blood vessels and to recanalization.

## 4. Rules of Use

### 4.1. Polymerization of Cyanoacrylate Glues

Cyanoacrylates are composed of one or more liquid monomers that harden quickly when they undergo polymerization. Polymerization of cyanoacrylates is anionic (i.e., induced by OH^-^ ions) and exothermic (i.e., accompanied by the release of heat, up to 80 °C). Cyanoacrylates harden in contact with fluids (which are rich in OH^-^ ions), such as blood in the vessel lumen [4]. Therefore, polymerization can occur too proximally (responsible for glue reflux and catheter adherence) or too distally. Cyanoacrylate should be injected through microcatheters with internal diameters not greater than 1 mm, and if possible under low pressure. The preparation should be sufficiently fluid to allow injection under these conditions. Polymerization starts nearly instantaneously. The cyanoacrylate glue must be injected slowly but decisively to avoid any reflux with occlusion of non-target vessels [14]. Polymerization starts when the mixture is in contact with blood. It starts 1 to 2 s after contact with blood and ends approximately between 60 and 90 s after contact. The speed of polymerization depends on the lipiodol/glue ratio but also on the volume and lasting of the previous flushing with dextrose solution 5% (D5) of the catheter and vascular bed (Figure 3). Table 1 recapitulates the mechanisms of action of cyanoacrylate glues. Common glue features are described in Table 2.

### 4.2. Dilution and Preparation of the Glue-Oil Mixture

The most widely used mixture is composed of 50% cyanoacrylate monomer and 50% ultrafluid ethiodized alcohol (Lipiodol^®^ Ultra Fluid, Guerbet). A 66% cyanoacrylate 33% Lipiodol^®^ mixture may be used for faster polymerization or a 33–66% mixture for slower polymerization. Radiological visibility of the glue-oil mixture may be improved by adding small amounts of a powdered heavy metal (usually tantalum). The mixture of cyanoacrylate, Lipiodol^®^, and tantalum must be prepared just before the injection.

When preparing the glue-oil mixture, great care must be taken to avoid contamination with ionic solutions, such as blood or normal saline, as this may accelerate polymerization and lead to excessively proximal glue deposition and/or an increased risk of catheter retention. A separate sterile table should be used to prepare the glue mixture just before the injection. Furthermore, the gloves should be changed before preparation then again before the injection. NBCA comes in small, sterile, 1 mL containers. Our preferred method to prepare the glue/Lipiodol^®^ mixture is aspiration of the glue out of its sealed container using a 1 mL syringe followed by glue injection into the desired volume of lipiodol in a 3 mL syringe, shot glass, or medicine cup. The two components are then thoroughly mixed (Figure 4).

Before transferring the prepared glue-oil mixture onto the field for injection into the patient, the hub of the microcatheter to be injected is placed on a clean, dry towel and flushed with D5. The catheter is then flushed thoroughly with D5 to completely clear ionic solutions from its lumen. This should be completed just before the glue injection. Different dilutions of glue and lipiodol can be used according to the goal of embolization, as shown in Table 3.

### 4.3. Radiological Visibility

On X-rays, a cast of the occluded vessel appears visible. By computed tomography (CT), high concentrations of tantalum or tungsten usually produce oversaturation artifacts. After contrast agent injection, the opacified vessels are undistinguishable from the occluded vessels. By magnetic resonance imaging, cyanoacrylate polymer alone produces a small signal void, whereas cyanoacrylate polymer combined with Lipiodol^®^ or tantalum generates a low- or iso-intensity signal. Polymerization results in hardening of the material, which forms a more or less complete glue cast of the occluded vessel lumen. Histological analysis shows incomplete occlusion (around 80%) of the lumen by the glue itself. Cyanoacrylate glue polymerization creates eddies that leave blood-filled interstices. The stasis and the heat released by polymerization leads to blood coagulation. About 15% of the vessel surface area is in contact with the glue.

### 4.4. Embolization Technique

Two methods can be used for cyanoacrylate injection: blocked- or controlled-flow injection to create vascular tree casts under pressure, and free-flow injection of small boluses of a mixture of cyanoacrylate and Lipiodol^®^ Ultra Fluid after highly selective catheterism to occlude shunts. The second technique requires smaller amounts of material.

We prefer 3 to 5 mL syringes to inject the glue-oil mixture. The syringe may be directly attached to the microcatheter hub to inject the desired volume of glue. Alternatively, a 3-way stopcock may be used to flush the catheter with D5 between pulses of glue-oil injection (called the sandwich technique). It can reduce the risk of microcatheter adherence and allows for control of the amount and depth of glue penetration. To prevent reflux around the microcatheter tip, which may attach the tip to the vessel wall (retained catheter), and polymerized glue withdrawal during catheter pullback, the mixture should not be injected too forcefully.

Among the factors that impact the depth of glue penetration, the most important are the flow rate through the catheter tip, the cyanoacrylate/Lipiodol^®^ ratio, and the injection technique. In vessels with rapid blood flow, greater control of the injection is needed. For this reason, a lower cyanoacrylate/Lipiodol^®^ ratio of 1:1 or 1:2 may be chosen to prevent penetration into the venous outflow and migration to the pulmonary arteries. This may occur during the treatment of arteriovenous malformations (AVMs) or in conditions involving arteriovenous shunting.

However, a more common technical problem is excessively proximal glue polymerization before penetration to the desired depth. A helpful method to achieve more distal embolization is the flood technique: a guide catheter in which the inner diameter is sufficiently larger than the outer microcatheter diameter is used to allow positive pressure continuous D5 flushing around the microcatheter. The resulting flushing of the distal vascular bed with a nonionic solution slows glue polymerization, thereby allowing deeper penetration of the mixture and more distal embolization. This technique is especially useful when embolizing a bleeding artery that is difficult to reach using the current generation of microcatheters. Once the injection is finished, the microcatheter is aspirated and rapidly withdrawn. The procedural overview is summarized in Table 4.

## 5. Potential Pitfalls

Most complications of glue embolization can be avoided by following the proper technique, which includes preventive measures. Early glue polymerization in the microcatheter leads to lumen occlusion and potential loss of vascular access. This is usually due to contamination with ionic fluids and may be avoided by appropriate flushing with D5 before the glue injection. If this occurs, changing the microcatheter is recommended, as attempts to clear the microcatheter lumen may result in uncontrolled expulsion of polymerized glue into a new and critical vascular branch, causing either proximal occlusion or nontarget embolization.

To avoid glue polymerization within the catheter, tip adherence, and microcatheter-lumen blockage, the catheter should be flushed with D5 solution to clear it of ionic material (contrast medium or blood) [9,15,16]. Then, the glue is injected and the microcatheter is rapidly removed to make sure that the tip does not adhere to the vessel.

Several techniques have been suggested for slowing the polymerization rate:Addition of a specific acid (acetic acid), which forms a conjugate with the activating anions;Addition of nitrocellulose;Addition of fluids, such as oily contrast agents, which are not directly involved in the polymerization process but form an emulsion with the monomer, thereby decreasing the three-dimensional profile of the glue polymer.

Excessive reflux around the microcatheter tip must be avoided. Reflux of polymerized glue around the microcatheter may cause tip adherence to the vessel wall, increasing the risk of microcatheter retention or nontarget embolization. If reflux is noted, the microcatheter tip must be pulled back sufficiently to free the tip before the glue polymerizes. Before the injection, redundant loops in the microcatheter should be removed to ensure that rapid and controlled pullback will be feasible. Exceptionally, the microcatheter tip may fracture during removal attempts. The tip is then usually left in place if it is not compromising flow to critical structures.

Glue mixture penetration into the venous circulation may lead to pulmonary embolism during the embolization of AVMs. Systemic embolism may occur in patients with right-to-left shunts, such as patent foramen ovale, for instance during the embolization procedure of AVMs or high-flow lesions. Although venous outflow blockage by glue in patients treated for intracranial AVMs may increase the risk of hemorrhage, it is not usually of clinical significance at peripheral sites. A higher glue concentration and proper injection technique can minimize this risk. The main advantages and drawbacks of cyanoacrylates are summarized in Table 5.

## 6. Clinical Applications

The clinical indications of endovascular NBCA injection include arterial embolization (peripheral AVMs, preoperative tumor shrinkage, hemostasis), venous embolization (of the portal vein, gonadal veins, or varices), and percutaneous embolization (e.g., for type 2 endoleaks or false aneurysms).

### 6.1. Arterial Embolization

#### 6.1.1. Peripheral Arteriovenous Malformations (AVMs)

AVMs are probably the main lesions in which cyanoacrylate glue embolization is beneficial, either by arterial approach or by direct puncture of the nidus. Indeed, AVMs are constituted by arteriovenous microfistulae through a vascular nidus. They are classified as central or peripheral, depending on whether they are located within the central nervous system or elsewhere. The ability of a liquid embolic agent to penetrate and occlude the nidus of the AVM is crucial (Figure 5). Because coils and detachable balloons occlude large vessels, they usually have no role in the embolization of complex AVMs located outside the lungs. Polyvinyl alcohol (PVA) particles have been used, but determining the appropriate particle size is difficult and the use of excessively small particles may result in pulmonary embolism [17]. Although absolute or 95% ethanol has been used, many interventionalists feel that ethanol carries an excessive risk of major complications [18]. These include injury to normal adjacent tissues, especially mucosal surfaces, skin, and nervous structures; and cardiopulmonary collapse related to the escape to the right cardiac chambers and pulmonary bed [19]. We reserve ethanol for organs in which the potential for nontarget tissue injury is minuscule, such as the kidneys, and for use as a sclerosing agent during direct puncture treatment of venous malformations. Because complex, multi-channeled, systemic AVMs are quite difficult to eradicate, treatment should be considered only in patients with significant symptomatology. Patients must be aware that the therapy is often palliative, and they should be prepared for the possibility of repeated sessions. A multidisciplinary approach with input from a variety of specialists is best. For lesions that are superficial or located at the extremities, it is important to work closely with plastic surgeons and vascular surgeons, especially when treating large abnormalities. Simple arteriovenous connections, as exemplified by classical acquired arteriovenous fistulas (AVFs), are generally best treated by selective embolization with a mechanical agent if this can be placed directly across the origin of the fistula and if the occluded arterial segment is not critical. The use of NBCA in large high-flow AVFs carries a significant risk of paradoxical embolization, although the initial delivery of coils to decrease the flow rate may diminish this risk [20]. When the lesion is too small to allow the placement of a catheter or microcatheter across the AVF, glue may allow successful occlusion.

#### 6.1.2. Tumors

Embolization may be used in selected cases as a preoperative treatment to reduce tumor-related pain, prevent tumor progression, induce tumor shrinkage, or stop acute tumor-related hemorrhage. Preoperative elective tumor embolization decreases blood loss and facilitates surgical resection [21,22]. Different types of hypervascular tumors are eligible for endovascular treatment. In the head and neck region, cyanoacrylate embolization is particularly indicated for meningioma, juvenile nasal angiofibroma, and paraganglioma [23,24]. At the spine, primary or secondary lesions that may benefit from selective embolization include aneurysmal bone cysts and metastases from renal cell carcinoma and thyroid cancer [25]. NBCA seems particularly beneficial in the treatment of asymptomatic and symptomatic renal angiomyolipoma as a means of shrinking the tumor and stopping inaugural bleeding (Figure 6) [26].

Glue embolization should be performed distally as close to the tumor as possible to maximize intratumoral penetration of the embolic agent. In some cases, direct tumor puncture is an alternative to endovascular embolization that obviates the need for multiple vessel catheterizations, which can be quite challenging.

#### 6.1.3. Hemostasis

Transcatheter arterial embolization (TAE) is now a major treatment for arterial bleeding. It has important advantages, such as effectiveness despite coagulopathy, which is often present due to anticoagulant medication or the effects of hemorrhagic shock [27]. Glue is administered faster compared to microcoils, a valuable asset especially in patients with hemodynamic instability [28]. Furthermore, the liquid consistency allows embolization of small or tortuous vessels. Preoperative contrast-enhanced CT is crucial for planning the procedure and identifying the artery responsible for the bleeding. The main expected CT findings are contrast extravasation or a false aneurysm-like lesion.

Bleeding sites accessible to TAE can be divided into three main territories: the viscera, the muscles, and the digestive tract.

##### Viscera

TAE with NBCA is a proven treatment of traumatic injuries to the liver, spleen, kidneys, and pelvis, with high success rates and few complications [29,30]. In 19 patients with bleeding due to pancreatitis or pancreatectomy, this procedure was consistently effective in stopping the bleeding with no complications, although three patients subsequently experienced recurrent bleeding. Glue may be particularly useful in postpartum hemorrhage with extravasation or pseudoaneurysm, as it may fill pseudo-aneurysmal sacs and provide more effective devascularization compared to a gelatin sponge [31].

##### Muscles

TAE with NBCA is efficient and safe for treating active intramuscular hemorrhage manifesting either as spontaneous hematomas in anticoagulated patients, which mainly involve the iliopsoas muscle and rectus sheath, or as traumatic or postoperative muscular bleeding [32,33].

##### Digestive Tract

TAE has been proven efficient in controlling hemorrhage and decreasing mortality when endoscopy cannot achieve hemostasis in patients with refractory upper gastrointestinal bleeding (UGIB) or lower gastrointestinal bleeding (LGIB). Many embolic agents can be used, such as coils, gelatin sponge, and particles. However, the choice of the best embolic material is still debated. Liquid adhesive agents, notably NBCA, have produced particularly good outcomes in both UGIB and LGIB. NBCA remains underused both overall and for TAE in gastrointestinal bleeding, due to concern about the risk of reflux during the procedure and nontarget embolization with potential ischemic bowel complications. However, a 2021 meta-analysis of 15 studies with 574 patients managed only with NBCA showed a low rate of bowel ischemia requiring surgery [34]. Indeed, whereas many interventional radiologists are apprehensive of using NBCA to treat gastrointestinal bleeding, this agent may be associated with lower major complication rates compared to other options (Figure 7).

### 6.2. Venous Embolization

#### 6.2.1. Portal Vein

Portal vein embolization (PVE) is highly recommended to promote liver growth before extensive hepatectomy in patients with liver cancer and an initially insufficient liver reserve. Evidence now exists that portal vein embolization with NBCA generates faster and greater liver regeneration compared with standard PVA particles and coils, allowing for earlier hepatectomy [35,36,37,38]. PVE with NBCA is also associated with smaller contrast volumes and shorter fluoroscopy times. This greater capacity compared to other embolic agents could be related to the provoked periportal inflammatory response [36]. NBCA allows for the best future remnant liver (FLR) hypertrophy. NBCA-lipiodol mixture can be used either by contralateral or ipsilateral approach. A high dilution (1:8 ratio) may be useful to obtain a distal embolization since the flow in the portal vein system is quite low. Particular attention should be devoted to avoiding FLR glue migration, since it can preclude liver surgery. When comparing complication rates between the groups with and without additional use of microcatheters, there were no cases of major NBCA migration or dislodgment with its use, and it was suggested that glue migration might be avoided with this coaxial technique. Indeed, as previously said, the use of a microcatheter is always welcome and preferable when using glue for PVE [37,38]. Meticulous attention is needed, especially near the end of the PVE procedure to avoid complications and glue migration.

#### 6.2.2. Gonadal Veins

Male varicocele is an extremely common condition that adversely affects quality of life and fertility. The incidence in young males is about 10–15% overall and can reach 45% in the subgroup with infertility [39,40]. Varicocele affects the pampiniform plexus and main spermatic vein trunks, which become abnormally dilated due to blood reflux toward the testicles. Treatment options are limited to percutaneous venous embolization and surgical therapy. Percutaneous embolization of the internal spermatic vein is associated with minimal pain and a quick recovery [41].

Varicocele embolization has been performed with many embolic agents, such as gelatin sponge particles, coils and microcoils, vascular plugs, detachable balloons, cyanoacrylates, and sclerosing agents [42,43,44,45]. The use of glue in this indication leads to less radiation exposure, less pain, and lower recurrence rates compared to other embolic materials [46]. Because NBCA and other liquid embolic agents have the advantage of penetrating into collateral pathways, they can be associated with greater effectiveness and a lower risk of recurrence (Figure 8) [47]. Glue embolization is also well indicated to embolize the “reservoir” in the case of symptomatic pelvic varices in females.

#### 6.2.3. Lower-Limb Varices

Endovenous thermal ablation (by laser or high frequency radio waves) proved safe and effective in patients with signs and symptoms related to great saphenous vein (GSV) incompetence [48]. Both techniques allow high venous occlusion rates but require perivenous tumescent anesthesia and may cause vein wall perforations, leading to postoperative pain and some other complications. Ultrasound-guided foam sclerotherapy is a chemical ablation technique that is gaining popularity, although repeat procedures are sometimes needed to maintain vein closure. Postprocedural inflammation and skin staining are common side effects. A general recommendation exists to prescribe support stockings after such venous interventions to facilitate vein closure and mitigate complications.

NBCA endovenous embolization holds considerable promise for treating lower-limb varicose syndrome. The first study of cyanoacrylate to treat saphenous vein incompetence in humans, reported in 2013, included 38 symptomatic patients and observed a 92% occlusion rate after 1 year [49]. Several studies have confirmed since then that glue embolization is feasible and provides extremely high GSV occlusion rates with very few mild-to-moderate side effects [50,51]. Moreover, the short procedure time and absence of a requirement for tumescent anesthesia and support stockings reduce patient discomfort compared to endovenous thermal ablation [52,53,54]. However, the incidence of postoperative phlebitis may be slightly higher than after endovenous thermal ablation, probably due to the greater inflammatory reaction induced by cyanoacrylate polymerization.

Further larger studies with long-term outcome assessments are required to identify the optimal therapeutic modalities for patients with great saphenous vein insufficiency.

### 6.3. Percutaneous Embolization

#### 6.3.1. Type 2 Endoleak

The most common type of endoleak is Type 2, which is defined as a retrograde flow into the excluded aneurysm sac via either the inferior mesenteric artery or lumbar arteries. The best treatment of patients with Type 2 endoleak remains debated. A recent study found no differences in outcomes between embolization by direct sac puncture (via the translumbar or transabdominal approach) and TAE [55]. A direct sac puncture has the advantage of allowing direct aneurysmal sac filling with a shorter procedure time and less radiation exposure (Figure 9). Various solid and liquid embolization materials have been used. When injecting NBCA alone or with coils, the glue fills the entire endoleak sac until it refluxes into the arterial afferences. Before NBCA injection, abdominal angiography must be performed to determine whether the inferior mesenteric artery (IMA) has an antegrade flow away from the sac. When this is the case, coils should be deployed at the origin of the IMA, or, if this is not possible, in the aneurysmal sac adjacent to the IMA origin, to prevent nontarget glue embolization of the IMA.

#### 6.3.2. False Aneurysms

Femoral artery pseudoaneurysm is a frequent complication after arterial puncture for endovascular interventions. Surgery has usually been considered the gold standard treatment. Nonsurgical options, such as ultrasound-guided compression, microcoil embolization, stent graft deployment, and percutaneous thrombin injection have been evaluated. Another potential option is percutaneous NBCA glue injection after ultrasound-guided puncture of the artery [56,57]. The main risk is distal glue spillage into the native artery through the neck of the false aneurysm. To prevent this, a balloon can be placed in front of the false aneurysm neck via contralateral femoral artery access before the injection of glue into the sac via ultrasound-guided puncture. Glue distribution is monitored by fluoroscopy until the sac is filled (Figure 10). This ultrasound-guided balloon-assisted glue embolization technique was successful in all 23 patients in a recent study [58].

## 7. Perspectives and Future Directions

### 7.1. Adding Ethanol to NBCA

The main concern with NBCA glue is its potential adhesion to the microcatheter and early polymerization within the catheter, leading to occlusion of the catheter lumen and incomplete embolization, as previously reported [58]. A basic science and swine study evaluated the effects of adding ethanol to NBCA and Lipiodo^®^. The mixture was less adhesive compared to NBCA-Lipiodol^®^ when the NBCA-Lipiodol^®^-ethanol (NLE) ratio was 1:1:2 or 1:1:3; the 1:1:1 ratio did not affect adhesion [59]. Adding ethanol accelerated and changed the configuration of NBCA polymerization via mechanisms that remain unelucidated. Thus, in swine, NLE formed a single large droplet with the 1:1:2 ratio and a noodle-shaped extrusion with the 1:1:3 ratio. NLE 1:1:2 or 1:1:3 may be injected in a controlled manner, without risking adhesion to the microcatheter, but must be prepared separately.

However, NBCA has been reported to induce acute vasculitis, and ethanol causes cell damage by deprivation of intracellular fluid and coagulation necrosis of the vascular wall and organ parenchyma [9,60,61,62,63]. Combining these two potentially highly irritating molecules may therefore be of concern. In swine, vessel wall changes were less marked with NLE (1:1:2 ratio) than with NBCA-Lipiodol^®^ (1:1 ratio) [64]. However, further evaluation of NLE is in order.

### 7.2. Aneurysm Remodeling with NLE

NLE may be useful for isolating and occluding an arterial lesion previously occluded by a balloon. For instance, the decreased adhesiveness allows balloon-assisted glue packing of wide-neck aneurysms [59,65,66]. Injected NLE becomes nearly solid, thereby producing highly effective aneurysmal occlusion. After balloon deflation, the microcatheter is navigated to the target, the balloon is re-inflated across the neck of the aneurysm, and the NLE is injected through the microcatheter. This technique produces complete packing of the aneurysmal sac. The microcatheter is moved in the same manner as for microcoil embolization with packing. A study in swine showed that the balloon catheter as well as the microcatheter could be easily removed after the glue injection [66]. Ethanol may increase the durability of the occlusion notably by promoting thrombogenesis in the aneurysm. According to another swine study, which used various NLE ratios, ethanol may act synergistically with NBCA-Lipiodol^®^ to promote sac occlusion and endothelium destruction, thereby preventing sac recanalization [65]. NBCA concentrations of at least 30% were required to avoid NLE migration. It should be noted that follow-up was only 3 days [65]. Other studies are needed to evaluate long-term sac occlusion and safety.

Another potential application of NLE may be in combination with detachable microcoils for aneurysmal sac packing to shorten the treatment time and reduce the number of microcoils, thereby limiting the cost. NLE may also hold promise for embolizing AVMs and controlling acute bleeding.

### 7.3. Prostate Artery Embolization

We demonstrated a new potential application of glue for prostate artery embolization (PAE) [67]. Indeed, PAE with NBCA is a feasible, safe, fast, and effective procedure with promising results in patients with benign prostate hypertrophy-related lower urinary tract symptoms. Further prospective comparative studies with longer follow-ups are required. However, the main advantage of using NBCA is the shorter procedural time compared to particulate embolization, which decreases the fluoroscopy time, and, therefore, the radiation dose to the patient. Another advantage of NBCA is that the fast polymerization from surface to core avoids the opening of pre-existing vascular anastomoses, an event reported with particles, thereby potentially decreasing the risk of nontarget embolization. NBCA/Lipiodol^®^ has many other advantages [67]. Lipiodol^®^ makes the embolic material radio-opaque, allowing for easier fluoroscopic guidance as compared as with other embolic agents that are not directly visualized, such as microparticles. Furthermore, NBCA is a fluid and may therefore be used to embolize vessels in which the microcatheter cannot be advanced, according to the blocked-flow technique. This situation is particularly frequent in PAE, as the flow rate in prostatic arteries is low. Furthermore, the lipiodol is taken up by the prostate gland, and the distribution of the treated prostate territories then becomes clearly visible by cone-beam computed tomography (CBCT) or CT. This distribution could be used as a surrogate marker of clinical success, as reported for the liver.

### 7.4. New Cyanoacrylates

Advances in research on liquid embolic agents can be expected to overcome some of the disadvantages of currently available products. Chemical mechanisms under study include polymerizing, precipitating, and phase-transitioning with the goal of obtaining solidified masses within the treated blood vessels. Inherent radiopacity is also being sought to avoid having to add radiopaque agents. Another goal is the reduction of streak artefacts on radiographs and CT scans. Solvent-free formulations that gel upon contact with blood are of interest as means of avoiding the injection of potentially toxic solvents [68].

NBCA-Lipiodol^®^ can be combined with the nonionic contrast agent iopamirone to form NLI. Iopamirone may decrease the irritant effect of the preparation on blood vessels, compared with ethanol. Furthermore, the mixture with iopamirone does not need to be prepared separately. When used to perform balloon-assisted embolization of 12 aneurysms in pigs, NLI of a 2:3:1 ratio was effective, with a similar decreased adhesiveness to that seen with NLE [69]. However, the histological effects of NLI and NLE were not compared in this study. Further research is needed to determine the durability of NLI aneurysm occlusion and to assess whether NLI promotes thrombogenesis within aneurysms [70].

## 8. Conclusions

Optimal endovascular treatment with NBCA glues requires considerable practical expertise with this embolic material. As with any other liquid embolic agent, a deep learning curve is needed. However, in specific indications, cyanoacrylate glues have many advantages. Although underused worldwide, cyanoacrylates can be particularly valuable in patients with hemodynamic instability due to bleeding, or in the case of coagulopathy, extreme vessel tortuosity, or narrowed vessels for which distal embolization with nonliquid embolic agents is impossible. Embolization with NBCA requires a careful evaluation of the vascular anatomy, close attention to technical details, and the use of Lipiodol^®^ oil to make the material radiopaque and to modulate the polymerization rate. Interventional radiologists should become more familiar with these embolic agents, which can be used as the first-line treatment for many peripheral abnormalities.

## Figures and Tables

**Figure 1 jcm-10-04320-f001:**
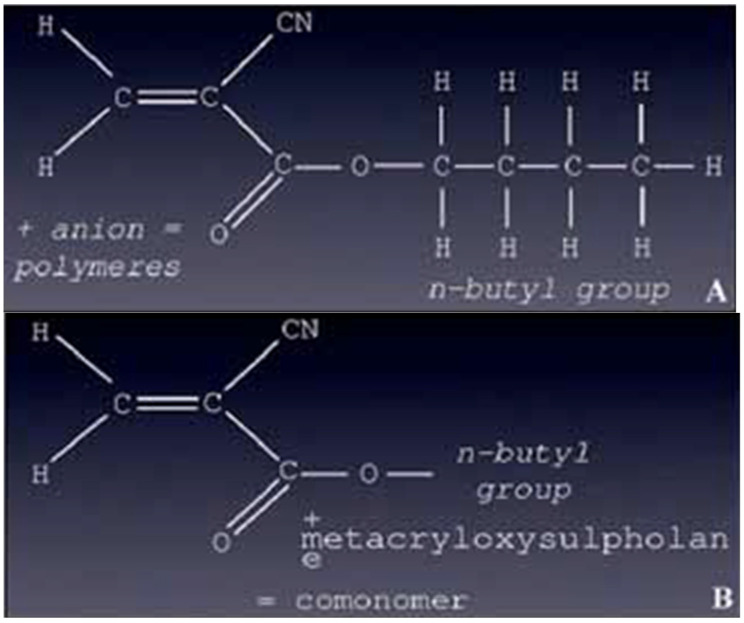
Chemical structure of two cyanoacrylate glues (*N*-butyl cyanoacrylate). (**A**) *N*-butyl-2-cyanoacrylate (NBCA) (Histoacryl^®^). (**B**) *N*-butyl-2-cyanoacrylate (NBCA) + metacryloxysulfolane (NBCA-MS) (Glubran^®^2).

**Figure 2 jcm-10-04320-f002:**
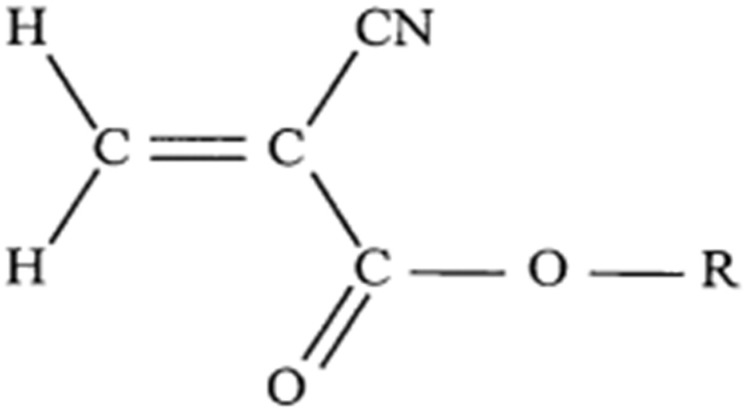
Chemical structure of monomeric cyanoacrylate. The R represents an alkyl group.

**Figure 3 jcm-10-04320-f003:**
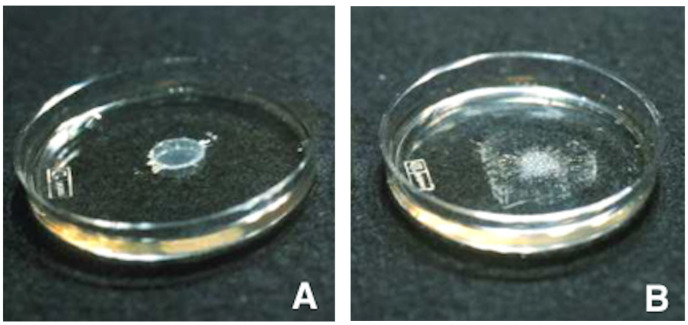
Example of extension of polymerization of glue according the lipiodol ratio. (**A**) Glue/lipiodol mixture in a 1:1 ratio (50% NBCA). (**B**) On the left glue/lipiodol mixture in a 1:4 ratio (20% NBCA) showing delayed polymerization with more extensive cast in surface.

**Figure 4 jcm-10-04320-f004:**
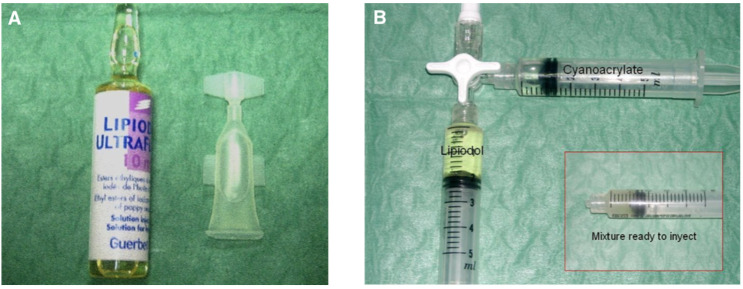
Features and preparation of glue and lipiodol as a mixture before embolization. (**A**) Lipiodol^®^ 10 mL vial and Glubran^®^2 1 mL vial. (**B**) Preparation of the glue/lipiodol mixture with 2 separated 5 mL plastic luer-locked syringes and a plastic 3-way stopcock in a smooth manner.

**Figure 5 jcm-10-04320-f005:**
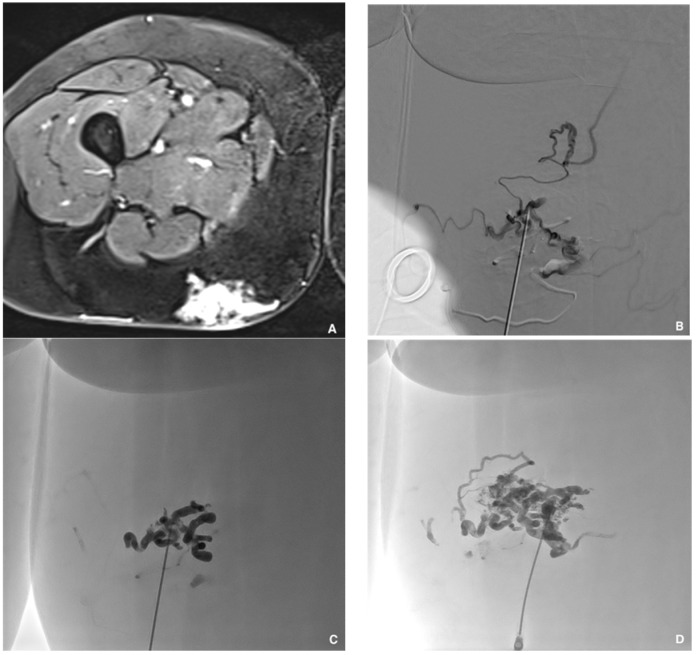
Superficial painful buttock arteriovenous malformation (AVM) in a 61-year-old patient. (**A**) Axial magnetic resonance imaging shows posterior subcutaneous contrast enhancement at the arterial phase, corresponding to the AVM. (**B**) Direct puncture of the nidus with a 21-gauge metallic needle and opacification of the nidus. (**C**) Embolization with a Glubran^®^2/Lipiodol^®^ mixture of a 1:5 ratio. (**D**) Final result after complete embolization of the nidus.

**Figure 6 jcm-10-04320-f006:**
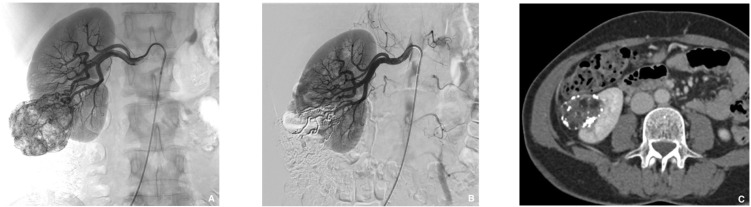
Prophylactic selective embolization of a large angiomyolipoma of the right kidney. (**A**) Angiography shows two feeding arteries to the angiomylipoma which is hypervascular. (**B**) Final control after selective and fast embolization of the two arterial branches with a Glubran^®^2/Lipiodol^®^ mixture of a 1:6 ratio. (**C**) Computed tomography scan at day 14 showing lipiodol uptake by the angiomyolipoma which is totally devascularized.

**Figure 7 jcm-10-04320-f007:**
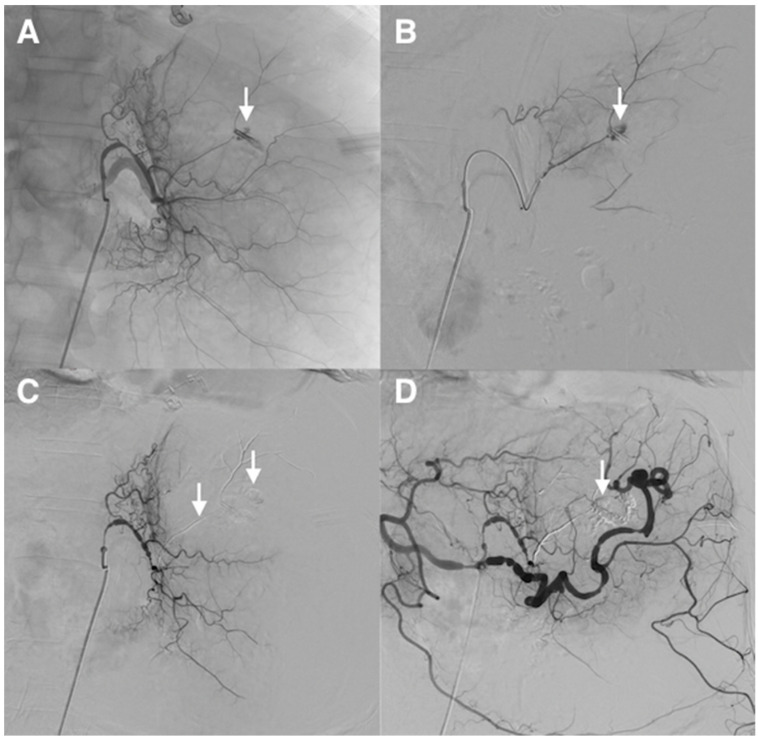
Refractory upper gastrointestinal hemorrhage from a gastric ulcer of the less curvature in a 64-year-old patient with coagulation disorders and hemodynamic instability. (**A**) Catheterism of the left gastric artery demonstrated extravasation of contrast medium from a small feeding branch of the less curvature (*arrow*). (**B**) Superselective angiogram of the bleeding artery, guided by an endoscopic metallic clip, confirmed active bleeding (*arrow*). (**C**) After selective microcatheterism, hemorrhage was controlled after superselective embolization of the feeding artery using a MagicGlue^®^/Lipiodol^®^ mixture (1:3 ratio) (*arrows*). (**D**) Final control angiography showed successful superselective occlusion of the bleeding arterial branch, respecting collaterals (*arrow*). The patient stopped to bleed immediately.

**Figure 8 jcm-10-04320-f008:**
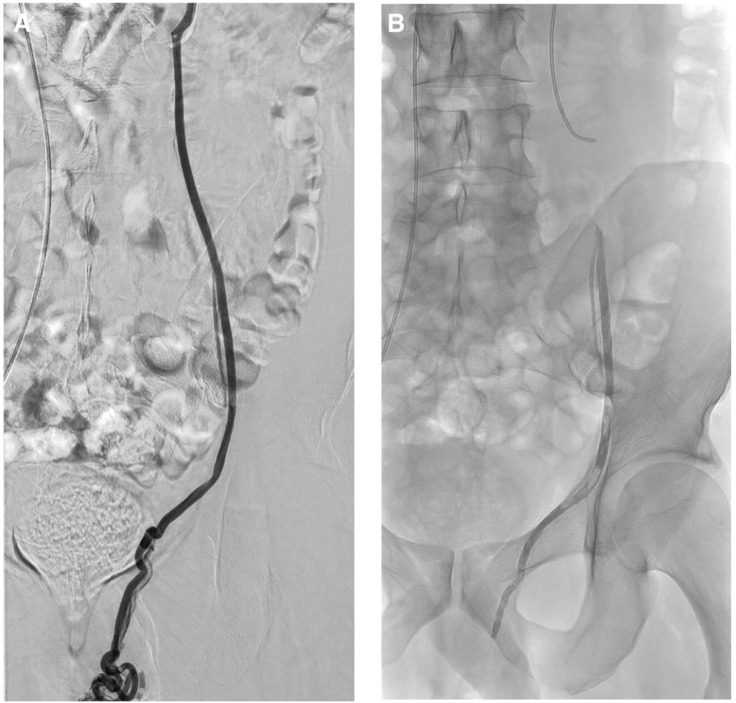
Typical left varicocele embolization with glue. (**A**) Left gonadal vein phlebography: a 5 French catheter is inserted in the left spermatic vein. A microcatheter is placed below the iliopectineal line and the dead space is filled with dextrose 5%, to avoid intracatheter glue polymerization. Embolization procedure is then performed using glue mixed with lipiodol at a ratio of 1:1 to get fast polymerization and avoid migration in case of reflux. Lipiodol–glue mixture is then injected under strict fluoroscopy, with continuous injection performed manually and a display of real-time distribution. The glue injection begins in the distal intrapelvic segment of the gonadal vein, and the catheter is withdrawn slowly while injecting NBCA under fluoroscopic control. Injection is then stopped before the pampiniform plexus is reached. The microcatheter is then removed when the glue fills the venous space selected beforehand. (**B**) Here we can see the cast of glue along the left spermatic vein after embolization.

**Figure 9 jcm-10-04320-f009:**
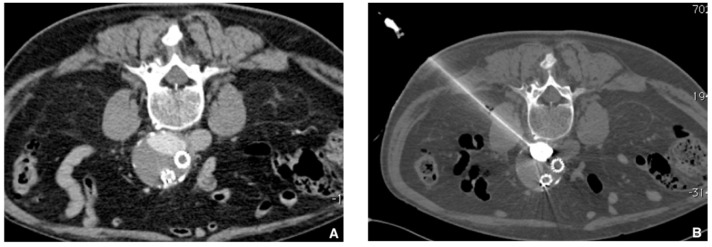
Type 2 endoleak from lumbar ateries. (**A**) Computer tomography (CT) scan with contrast injection in procubitus position showing posterior opacification of the periprothetic aneusrysmal sac (nidus). (**B**) Translumbar puncture of the nidus of the endoleak under CT guidance with a metallic needle. Insertion of a microcatheter inside and embolization of the nidus with a Glubran^®^2/lipiodol mixture (1:4 ratio) until complete nidus opacification.

**Figure 10 jcm-10-04320-f010:**
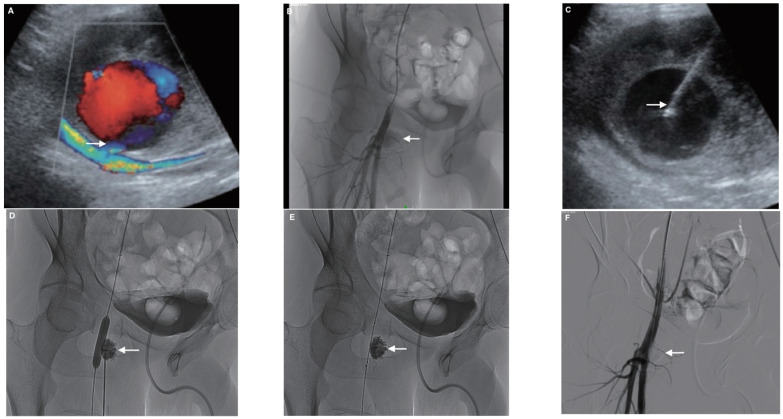
Example of a 62-year-old patient who developed a painful pulsatile mass with groin hematoma following coronary angioplasty. (**A**) Duplex ultrasound (US) at the groin puncture site demonstrated a large false aneurysm (FA) arising from the right proximal superficial femoral artery. (**B**) Angiography by crossover from the left side confirmed the FA from the superficial femoral artery (arrow). (**C**) The FA sac was then punctured under US guidance with a metallic needle with hyperechoic tip (arrow). (**D**) After inflating a Mustang^TM^ balloon of appropriate size in the parent artery in front of the FA neck in order to avoid reflux, a Glubran^®^2/lipiodol of a ratio of 1:1 was injected into the FA sac under fluoroscopic guidance until complete filling of the FA (arrow). (**E**) Arteriogram after deflating the balloon demonstrated, few minutes after glue injection, no NBCA migration (arrow). (**F**) Final angiography after embolization showed complete occlusion of the FA (arrow) and confirmed patency of the parent artery.

**Table 1 jcm-10-04320-t001:** Cyanoacrylate glues: types and mechanisms of action.

Characteristics	Methyl	Butyl		Hexyl
**Type**	NMCA	NBCA	NBCA + CM	NECA
Name	SuperGlue^®^	Histoacryl^®^ Trufill^®^	Glubran^®^2	MagicGlue^®^
CH_3_ radical	+	++	++	+++
Polymerization time	Very fast	Fast	Intermediate	Low
Cytotoxicity	+++	++	+	+
Inflammatory reaction	+++	++	+	+
Adhesive strength	+++	+++	++	+

CH_3_, methyl; NMCA, *N*-methyl cyanoacrylate; NBCA, *N*-butyl cyanoacrylate; CM, comonomer; NECA, *N*-hexyl cyanoacrylate; methyl glue is not for endovascular purposes whereas butyl and hexyl glues are used for endovascular purposes; +++, important; ++, moderate; +, low.

**Table 2 jcm-10-04320-t002:** Common cyanoacrylate glue features.

Characteristics
Allows quick embolization
Permanent embolic agent
Clinical efficacy does not depend on coagulation parameters
May reach distal target that cannot be navigated with microcatheters
Highly adhesive
Hemostatic and sclerosant
Bacteriostatic
Can dissolve polycarbonates
Typical smell
Cold storage needed

**Table 3 jcm-10-04320-t003:** Choosing glue/lipiodol dilution according to the situation.

Variables	Low Dilution (1:1–1:3)	High Dilution (1:4–1:9)
Catheter position	Close to lesion	Away from lesion
Catheter tip	Wedged	Free
Injection manner	Continuous column	Drop by drop
Flow speed	Very fast	Slower
Occlusion	More proximal ±	More distal ±
Application	High flow	Intermediate flow

±, more or less.

**Table 4 jcm-10-04320-t004:** Tips and tricks overview for cyanoacrylate procedure embolization.

Tips and Tricks
Local anesthesia
Accurate identification of anatomy/feeders
Use of a coaxial technique
Microcatheter needed
Previous flushing with dextrose 5% of the dead space
Mixing with Lipiodol^®^
Make the mixture radiopaque
Modulate the rate of polymerization
Use of a plastic 3-way stopcock
Use of non-polycarbonate luer-locked syringes
Slow and regular injection of the mixture under fluoroscopy
Wedged
Free-flow injection
One shot or multi shots
Blocked flow injection
Do not rush with microcatheter withdraw
Pull out curtly the microcatheter after getting the goal
Avoid doing ‘that little bit more’ just to make the final result look better
Flushing the microcatheter with dextrose 5% for reuse

**Table 5 jcm-10-04320-t005:** Advantages and drawbacks of cyanoacrylate glues.

Advantages	Drawbacks
Fast polymerization allows for less procedural radiation dose	May stick to the catheter if not used properly
Permanent occlusion prevents recanalization	Very adhesive with inflammatory reaction
Efficacy does not depend on coagulation parameters	Must be mixed with lipiodol to be radiopaque
Can reach distal targets that cannot be navigated with microcatheters	Learning curve by the operator is needed to use it
Cheap *	Use of a microcatheter is necessary

* In Europe, not in the USA.

## Data Availability

All the study data are reported in this article.

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
