# Peer review of "Endovascular Use of Cyanoacrylate-Lipiodol Mixture for Peripheral Embolization: Properties, Techniques, Pitfalls, and Applications"

_jcm, 2021, doi:10.3390/jcm10194320_

Round 1
Reviewer 1 Report
I congratulate the authors for the work done and to consider publishing “Review Endovascular Use of Cyanoacrylate-Lipiodol Mixture for Peripheral Embolization: Properties, Techniques, Pitfalls and Applications ” in JCM.
Specific comments
Improve the quality of figures especially figures 3 and 4.
page 4-18
line 130: should be cited as Solution of Dextrose 5% (D5) or Glucose 5% (G5). This has not been cited before but cited later.
Table-4: should be organized differently because it is confusing. Does the preparation have anything to do with the tips and tricks?
- Potential Pitfalls
Don't understand why PAE is included in Potential pitfalls and neither reference shows this promising results with NBCA for PAE.
"PAE with NBCA is a feasible, safe, fast, and effective procedure with promising results in patients with BPH-related LUTS. Further prospective comparative studies with longer follow-ups are warranted to identify the best embolic agent and the patients most likely to benefit from PAE to treat BPH-related LUTS." (This should be placed elsewhere with references).
Table-5
Cheap in Europe, not in the US.
Figure-8 should include a venogram before embolization. Embolization procedure and final image after glue embolization.
lower limb varices
Should include venaseal meta-analysis "Kolluri R, Chung J, Kim S, Nath N, Bhalla BB, Jain T, Zygmunt J, Davies A. Network meta-analysis to compare VenaSeal with other superficial venous therapies for chronic venous insufficiency. J Vasc Surg Venous Lymphat Disord. 2020 May;8(3):472-481.e3. doi: 10.1016/j.jvsv.2019.12.061. Epub 2020 Feb 14. PMID: 32063522."
Author Response
Responses to Reviewer 1 Comments
I congratulate the authors for the work done and to consider publishing “Review Endovascular Use of Cyanoacrylate-Lipiodol Mixture for Peripheral Embolization: Properties, Techniques, Pitfalls and Applications” in JCM.
Reply: Thank you very much for your comments.
Specific comments
Improve the quality of figures especially figures 3 and 4.
Reply: Thank you very much for your comments. The quality of figures 3 and 4 has been improved as suggested. The quality of all other images is quite good.
Page 4-18: line 130: should be cited as Solution of Dextrose 5% (D5) or Glucose 5% (G5). This has not been cited before but cited later.
Reply: Thank you very much for your comments. It has been corrected and first described as dextrose solution 5% before using the abbreviation D5, as suggested.
Table-4: should be organized differently because it is confusing. Does the preparation have anything to do with the tips and tricks?
Reply: Thank you very much for your comments. We fully agree. The table has been reorganized for more understanding. All sentences have been included in a tips & tricks section only. The word “preparation” has been removed.
Potential Pitfalls
Don't understand why PAE is included in Potential pitfalls and neither reference shows this promising results with NBCA for PAE.
"PAE with NBCA is a feasible, safe, fast, and effective procedure with promising results in patients with BPH-related LUTS. Further prospective comparative studies with longer follow-ups are warranted to identify the best embolic agent and the patients most likely to benefit from PAE to treat BPH-related LUTS." (This should be placed elsewhere with references).
Reply: Thank you very much for your comments. We fully agree. This is a mistake. This paragraph is already included in the PAE paragraph in the “perspectives and future directions” section as a new potential application. Then, it has just been deleted from this pitfalls section.
Table-5
Cheap in Europe, not in the US.
Reply: Thank you very much for your comments. It has been added in the footnote of the Table 5 as suggested.
Figure-8 should include a venogram before embolization. Embolization procedure and final image after glue embolization.
Reply: Thank you very much for your comments. The figure 8 has been improved with venogram before embolization and final image after glue embolization, as suggested.
Lower limb varices
Should include venaseal meta-analysis "Kolluri R, Chung J, Kim S, Nath N, Bhalla BB, Jain T, Zygmunt J, Davies A. Network meta-analysis to compare VenaSeal with other superficial venous therapies for chronic venous insufficiency. J Vasc Surg Venous Lymphat Disord. 2020 May;8(3):472-481.e3. doi: 10.1016/j.jvsv.2019.12.061. Epub 2020 Feb 14. PMID: 32063522."
Reply: Thank you very much for your comments. This reference has been added as ref 51, as suggested. All references have been renumbered.
Reviewer 2 Report
The article is well constructed and exhaustive on the applications of the NBCA.
The subdivision of the paragraphs is clear and complete. The authors elaborated in a detailed and complete way the different and multiple applications of the NBCA outlining the advantages, the related risks and the innovations of the research.
Pag 6, lines 214-217 this pragraph about the PAE is out of place and should be included in the clinical applications part of the NBCA.
Author Response
Responses to Reviewer 2 Comments
The article is well constructed and exhaustive on the applications of the NBCA.
The subdivision of the paragraphs is clear and complete. The authors elaborated in a detailed and complete way the different and multiple applications of the NBCA outlining the advantages, the related risks and the innovations of the research.
Reply: Thank you very much for your comments.
Page 6, lines 214-217 this paragraph about the PAE is out of place and should be included in the clinical applications part of the NBCA.
Reply: Thank you very much for your comments. I think there is a mistake. Lines 214 to 217 on page 6 are not focused on PAE at all. Then no changes have bene made in this paragraph. On the other hand, the paragraph on PAE in the pitfalls section has been deleted because out of place and is already included in the “perspective and future directions” section as a new and fresh potential application.